# Navigating Like a Fly: *Drosophila melanogaster* as a Model to Explore the Contribution of Serotonergic Neurotransmission to Spatial Navigation

**DOI:** 10.3390/ijms24054407

**Published:** 2023-02-23

**Authors:** Ivana Gajardo, Simón Guerra, Jorge M. Campusano

**Affiliations:** 1Departamento de Biología Celular y Molecular, Facultad de Ciencias Biológicas, Pontificia Universidad Católica de Chile, Santiago 8331150, Chile; 2Departamento de Neurociencia, Instituto Milenio de Neurociencia Biomédica (BNI), Facultad de Medicina, Universidad de Chile, Santiago 8380453, Chile

**Keywords:** serotonin, *Drosophila*, spatial navigation, spatial memories

## Abstract

Serotonin is a monoamine that acts in vertebrates and invertebrates as a modulator promoting changes in the structure and activity of brain areas relevant to animal behavior, ranging from sensory perception to learning and memory. Whether serotonin contributes in *Drosophila* to human-like cognitive abilities, including spatial navigation, is an issue little studied. Like in vertebrates, the serotonergic system in *Drosophila* is heterogeneous, meaning that distinct serotonergic neurons/circuits innervate specific fly brain regions to modulate precise behaviors. Here we review the literature that supports that serotonergic pathways modify different aspects underlying the formation of navigational memories in *Drosophila*.

## 1. Introduction

Neuromodulation confers flexibility to anatomically restricted neural networks so that animals can adequately respond to complex external demands [1]. Serotonin (5-hydroxytryptamine, 5-HT) is a neuroactive molecule that modulates behavioral functions that are planned and executed via circuits in the central nervous system (CNS) in both invertebrates and vertebrates. Serotonergic neurotransmission promotes changes in the structure and activity of brain areas relevant to cognitive abilities, such as spatial navigation, which is evolutionarily conserved [2,3]. The serotonergic system exhibits common organizational principles in vertebrates and invertebrates, such as similar biosynthetic and reuptake pathways and a relatively small number of serotonin-releasing neurons compared with the total number of brain neurons. Moreover, serotonin exerts modulatory roles by a wide distribution of projections reaching numerous neural targets in the CNS of both vertebrate and invertebrate animals [4,5,6].

Genetic tools generated over the years in *Drosophila melanogaster* (vinegar fly) have helped identify discrete serotonergic neuronal subpopulations and circuits that modulate behaviors at different levels of complexity [7,8,9,10,11,12,13,14]. Compared to other animal models, the fly offers experimental advantages in assessing whether distinct serotonergic neurons modulate specific neuronal circuits underlying cognitive abilities, such as spatial navigation. Here, we will review the evidence that supports the contribution of serotonin to different behavioral aspects of spatial navigation, such as sensorimotor responses and spatial memory processing.

## 2. A Brief Description of the Serotonergic System in *Drosophila melanogaster*

Serotonin is considered both a classic neurotransmitter and also a neuromodulator [15]. In vertebrates and invertebrates, neuromodulation occurs when serotonin is released extrasynaptically from the soma and dendrites, and from axon varicosities in synapses (Figure 1). In that sense, serotonergic neurons do not necessarily need to form classical synapses to communicate with another neuron and modulate its activity [16,17]. Below, we describe the serotonergic system in *Drosophila* at the molecular and anatomical levels.

### 2.1. Molecular Organization of Serotonergic Components in the Adult Fly

Like other aminergic systems, the serotonergic system and its cellular components are highly conserved between vertebrates and invertebrates (Figure 1). Thus, the genome of mammals and *Drosophila* contain genes encoding presynaptic cellular components. These include the biosynthetic enzymes tryptophan hydroxylase, dTRH [18] and dopa decarboxylase, dDDC, which is also known as Aromatic L-amino acid decarboxylase [19,20], the vesicular monoamine transporter dVMAT [21], and also the plasma membrane serotonin transporter dSERT, which is responsible for amine transport back into the presynaptic terminal [22]. Notably, the *Drosophila* genome does not seem to contain a gene encoding for the monoamine oxidase, MAO, which in vertebrates is an essential enzyme responsible for the metabolization of amines [23,24].

As in vertebrates, the synthesis of serotonin in flies implicates the action of the enzyme tryptophan hydroxylase (dTRH) [18] and the enzyme dopa decarboxylase (dDDC), which is common for dopamine synthesis [5,19]. The activity of the enzyme tryptophan hydroxylase is rate-limiting in serotonin biosynthesis. In *Drosophila*, two enzymes exhibit tryptophan hydroxylase activity: one, responsible for serotonin synthesis in neurons, known as dTRHn, while a different enzyme produces serotonin in non-neuronal tissues, the phenylalanine hydroxylase (dTPHu) [5,25]. Thus, dTRHu and dTRHn resemble TPH1 and THP2 in vertebrates. The expression of dTRH and dDDC is regulated in a developmental-stage-specific manner throughout the life cycle of *Drosophila* [26].

The transport of serotonin across compartments occurs via dVMAT and dSERT. First, serotonin is stored in synaptic vesicles by the action of the vesicular monoamine transporter (VMAT). dVMAT is a member of the SLC18 subfamily, and in *Drosophila,* two variants have been described: dVMAT-A and dVMAT-B. The serotonergic neurons express the dVMAT-A variant [21,27]. On the other hand, serotonin reuptake from the extracellular environment occurs by the action of dSERT, which is homologous to the human and rodent SERT (hSERT and rSERT). This transporter binds serotonin with high affinity as its human orthologue [28]. dSERT exhibits similar pharmacological properties when compared to vertebrates; for instance, its activity can be inhibited by a variety of chemicals, including fluoxetine, which is a potent SERT blocker [29]. In adult flies, dSERT is widely expressed throughout the brain and is found in cellular bodies, arborizations, and varicosities [30]. Moreover, all the serotonin-containing somas are also positive for dSERT expression and vice versa [30].

Five metabotropic receptors for 5-HT have been described in *Drosophila*: d5-HT1A, d5-HT1B (expressed both pre- and postsynaptically), d5-HT2A, d5-HT2B y d5-HT7 [31,32,33]. At least three of these receptors mediate excitatory events in postsynaptic neurons, while one receptor is inhibitory [34]. In particular, d5-HT1A and -1B receptors are orthologous to mammalian 5-HT1A, which is found pre- and post-synaptically, and has been associated with several behavioral disorders [35,36]. There is evidence showing that d5-HT1B can act as an autoreceptor [5,37,38], while d5-HT1A is located presinaptically [39], which supports the idea that it could also act as an autoreceptor by analogy to its mammalian orthologous 5-HT1A. In addition, both are coupled to inhibitory Gi/Go, reducing the level of cytosolic cAMP due to the inhibition of the adenylyl cyclase (AC) activity [40,41]. The d5-HT2A and d5-HT2B receptors (ortholog to mammalian 5-HT2) are coupled to Gq, and their activation leads to an increase in calcium levels via activation of the phospholipase C (PLC) pathway [42,43,44]. The d5-HT7 receptor activates the Gs protein, which stimulates the AC enzyme, increasing cytosolic cAMP [45]. 

### 2.2. Anatomical Organization of Serotonin-Releasing Neurons and Their Projections in the Adult Brain

Although the brains of vertebrates and *Drosophila* exhibit some obvious anatomical differences, the organization of the serotonergic systems in these animals share some common basic principles. The different clusters of serotonergic neurons were discovered in rodents [46] by the so-called Falck–Hillarp method, which uses formaldehyde vapor to generate a yellow fluorescent signal in serotonergic neurons [47]. The brains of humans, rats, and mice contain about 300,000, 30,000, and 26,000 serotonin-releasing neurons, respectively [48,49,50], which are organized into nine clusters (B1–B9). The somas are distributed in a caudal to rostral organization in the raphe nuclei and send their projections to anterior or posterior areas of the vertebrate brain [2,51]. B1–B3 clusters form the caudal group in the medulla and project their axons to the spinal cord and the periphery. B4–B9 clusters form the rostral group from the pons to the midbrain, subdivided into the dorsal raphe group (B6 and B7) and the median raphe group (B5 and B8) [6,52]. The alphanumeric nomenclature refers only to the order of appearance of serotonergic clusters in brainstem sections.

Comparably, *Drosophila* researchers have identified serotonergic neuron clusters in the adult fly brain by immunostaining using anti-serotonin or anti-Trh antibodies [12,53]. By using this strategy, it has been determined that the adult fly brain contains ~100 serotonin-releasing neurons, and that 11 neuronal populations can be distinguished per hemisphere, out of which the AMP and ADMP have only one neuron per cluster (see Figure 2 and Table 1) [12,30,53,54]. The Gal4/UAS system [55,56] has been used to drive the expression of fluorescent proteins and complement the results obtained by immunostaining methods. The GAL4/UAS system combines the cloning of a transgene downstream of a UAS binding site (upstream activation sequence) and the use of a specific cell type promoter. The Trh promoter is commonly used to drive the expression of the Gal4 transcription factor in serotonergic neurons. Then, GAL4 expressed in serotonergic neurons directs the expression of a given transgene, such as green fluorescent protein (GFP). Table 1 shows that using Trh-Gal4 it is possible to recognize fewer serotonergic neurons than those stained with the anti-serotonin antibody, and that other neuronal populations are also identified [12]. Importantly, by using these approaches it has been shown that—similar to other monoaminergic populations—the somas of serotonergic neurons are distributed in the adult brain periphery, and they project towards the brain center where the neuropils are found [57].

Over the years, the nomenclature of serotonergic clusters has changed. The changes are illustrated in Table 1. Vallés and White [53] reported the existence of nine noticeable groups distributed in the entire adult brain using immunostaining against serotonin. The clusters described by Vallés and White [53] were named lateral protocerebrum (LP1 and LP2), supraesophageal ganglion (SP1 and SP2), ipsilateral IP, and subesophageal ganglion (SE1, SE2, and SE3). Sitaraman et al. [59] identified more clusters in the brain’s anterior region, which were named anterior medial protocerebrum (AMP) and anterior lateral protocerebrum (ALP). In addition, some of the previously described clusters were redefined to propose the existence of the posterior medial protocerebrum (PMP) and posterior lateral protocerebrum (PLP) clusters [10,54]. Later, Pooryasin and Fiala [12] proposed that SE1 and SE2 clusters, which are located in the anterior area of the brain, could be renamed lateral subesophageal ganglion (SEL). In contrast, the SE3 cluster could be called medial subesophageal ganglion (SEM), and the LP2 cluster could change its name to lateral protocerebrum (LP). Moreover, Pooryasin and Fiala [12] described a new cluster localized in the anterior dorsomedial proto-cerebrum, which was named anterior dorsal-medial paired somata (ADMP), and further proposed the division of the PMP cluster into three new groups, PMP-dorsal, -medial, and -ventral. This new redefinition of clusters acknowledges better the distribution of the different neuronal populations in the fly brain. Importantly, it is consistent with more recent reports which studied the organization of serotonergic clusters in the fly brain using TRH-Gal4-driven GFP expression [60], and an antibody staining against TRH [58] (see Figure 2). 

The complementary use of immunostaining and the Gal4/UAS system have made it possible to identify the wide distribution of serotonergic projections in the fly brain. However, better characterization of serotonergic projections has been obtained using sophisticated tools, such as the stochastic approach FRT-FLP system [11,12], and the electron microscopy-based *Drosophila* brain connectome that allows reconstruction of individual neurons [61,62]. 

Most neuropils of the central fly brain and in the optic lobes (OL) show innervation of 5-HT-immunoreactive fibers [39,60]. For instance, the lateral horn (LH), a dorsal neuropil, is innervated by fibers from AMP [63,64] and PMPV [12] clusters. The antennal lobes (AL), which are the fly brain structure homologous to the vertebrate olfactory bulb, are innervated by fibers from the serotonergic AMP [63,64,65] and PLP clusters [11,64,65]. Lateral neuropils located in the OL, which are associated with the processing of visual information, receive axons from the LP/PMPV cluster [12,39]. A very important neuropil in the insect brain that receives and processes olfactory information, the Mushroom Bodies (MB), receives serotonergic projections at different levels [11,64,66,67]. Another important neuropil in the fly brain is the central complex, which has been associated with the execution of motor programs. It is possible to find 5-HT-immunoreactive fibers in this *Drosophila* brain region arising from PLP and PMPD clusters, which innervate the dorsal portion of the so-called fan-shaped body (FB) and the ellipsoid body (EB), respectively [11,12,62].

Here below, we will review the information that supports that serotonin is able to modulate the activity of neurons constituting fly brain neuropils associated with spatial navigation.

## 3. Serotonergic Neural Circuits on Different Aspects of Spatial Navigation

In *Drosophila*, expression binary systems (Gal4/UAS, LexA-LexAop, or Q system) [68] and advanced genetic tools [69] which include newly generated Gal4 lines such as those from the Janelia collection, Split Gal4, FLP-FRT, and CRISPR/Cas systems [70,71,72], have helped elucidate the involvement of defined serotonergic neurons and specific neural circuits in behaviors. Thus, it has been assessed the consequences of the structural and anatomical heterogeneity of serotonergic neurons in behaviors such as motor activity and locomotion [9,12], sleep and circadian rhythm [7,73,74], social interactions [75] and aggressivity [11,76,77,78], courtship and mating [12,31,79], depressive states [8], and learning and memory [7,13,54]. In contrast, the modulatory role of serotonin-releasing neurons in neuronal circuits underlying cognitive abilities such as spatial navigation, has been little reviewed. 

In vertebrates and invertebrates, the ability to navigate and maintain a sense of location while integrating information from complex environments is a fundamental cognitive function critical to survival. To move in complex environments, animals use navigational strategies involving sensorimotor responses and memory processing of spatial and non-spatial information, including sensory cues such as food odors, temperature and visual stimuli (Figure 3A). 

Among the behavioral features associated with spatial cognition, spatial navigation is one of the most complex skills, which is evidenced by the intricate neural networks involved in this behavior. In vertebrates, spatial navigation involves the medial entorhinal cortex (MEC), the hippocampal region and projections from the medial septum connections [80]. The cholinergic, GABAergic and glutamatergic neurons in the medial septum send their axons to grid and place cells in the MEC and the hippocampus. These are some of the neurons whose activity depends on the navigational strategies followed by an animal [80,81,82]. Serotonergic neurons modulate these connections. For instance, it has been shown that projections from neurons of the B8 cluster innervate hippocampal neurons that express 5-HT1A, 5-HT1B, and 5-HT7 receptors [83]. Consistently, KO mice for different 5-HT receptors are deficient in different aspects of spatial learning [84,85,86].

In flies, the neuronal circuits underlying visually guided spatial navigation involve brain areas receiving and processing sensory information of different qualities and modalities (Figure 3B). However, some of the fly brain regions most associated with planning and executing spatial navigation are the OL, the EB and the protocerebral bridge (PB) (reviewed in [87]). Additionally, spatial navigation comprises cells with different neurochemical identity such as inhibitory GABAergic and excitatory cholinergic neurons [62,88], and also neurons containing and releasing neuromodulators. 

In *Drosophila*, serotonin modulates different behavioral responses to multiple sensorimotor modalities involved in spatial navigation. However, the serotonin influence on sensory processing and then the execution of behaviors is complex, which is explained by the heterogeneous characteristics of the serotonergic circuits and their target areas. This is exemplified when analyzing one specific attribute of fly movement, the walking speed, which is the sum of the contribution of serotonin’s action on specific receptors expressed in various brain locomotor areas [9].

Flexible locomotor behavior is necessary to navigate complex environments, where the walking speed must be tailored to the needs of challenging terrains. In flies, different aspects of walking are orchestrated by the neural circuits located in the ventral nerve cord (VNC), which has three pairs of thoracic neuromeres (T1, T2, and T3), coordinating the movements of corresponding pairs of legs [89]. The VNC is an analog of the vertebrate spinal cord and receives serotonergic signaling. Accordingly, the activation of serotonergic neurons that release serotonin in the VNC causes flies to walk slower but maintain coordination. The silencing of these neurons causes flies to walk faster in complex contexts. In addition, an increase in walking speed is observed in mutant flies for the d5-HT7 receptor, suggesting that this receptor is responsible for mediating the effects of serotonin on walking speed [9]. Comparably, Pooryasin and Fiala [12] designed a stochastic targeting of serotonin neurons that mediates specific aspects of locomotor inactivity. In this study, two pairs of serotonin neurons from the PMPV cluster are sufficient to induce behavioral quiescence—defined by a decreased locomotor activity—but do not exert their effects on all aspects of this behavior [12]. These antecedents support that distinct serotonergic circuits underlie many aspects of locomotor behavior.

Memory processing of non-spatial information obtained from sensory cues such as food odor is critical to navigation. In flies, odor perception is suppressed by the stimulation of four distinct serotonergic neurons from the posterior protocerebrum [90]. In the adult brain, serotonergic neurons reach several brain regions associated with the processing of olfactory information, including the AL, LH, and MB [1,8,64,91]. These regions express different subtypes of serotonergic receptors (d5-HT1, d5-HT2, and d5-HT7) and receive innervation from a discrete number of serotonergic neurons [53], including the CSD cluster (also named AMP) [30,92]. Evidence has demonstrated that inhibiting neurotransmission of all serotonergic neurons results in a strong deficit in short-term appetitive olfactory memory [54]. Interestingly, the specific activation of a group of 25 serotonergic neurons from ALP, SEL, LP, PLP, PMPV, and PMPD clusters significantly increases midterm memory performance [7]. Moreover, silencing neurotransmission of a pair of serotonin-releasing neurons identified as SPN, which seem to be part of the SEL cluster, decreased long-term memory performance. Silencing neurotransmission of this specific cluster reduced short-term olfactory memory [14]. Contrarily, silencing transmission of the same neuronal population did not affect long-term anesthesia-resistant memory performance [14]. All these results support that serotonergic neurons in specific clusters modulate the generation of different forms of olfactory memories.

The temperature is a non-visual environmental factor that drives animals to specific locations. In flies, cells in the antennae sense the environmental temperature. The early thermosensory processing involves the AL and LH, while in the MB more complex integration in temperature memory or preference occurs [93]. In this sense, the assessment of non-visual place learning has been studied using high temperature as a negative reinforcer in a spatial operant learning paradigm named heat-box [94]. Sitaraman et al. [59], using high temperature as a negative reinforcer, demonstrated that inhibition of the activity of serotoninergic neurons reduced non-visual place memory performance. Additionally, activating neurotransmission in all serotonergic neurons during the training session increased performance. This behavioral response is associated with SEL, SEM, and PMPD clusters [13]. In a different work, Sitaraman et al. [13] evaluated the time it takes the flies to move to a safe location when they are exposed to the heat punishment. When all serotonergic neurons were activated, an increase in escape latency was observed, even with repetitive punishments. Contrarily, inhibiting the same neurons decreases the escape latency. These antecedents suggest that serotonergic neurons modulate different aspects of the non-visual place memory associated with high temperature. In doing this, serotonergic neurons innervate several brain neuropils associated with the processing of temperature information and memory.

Vision plays a critical role in navigation in vertebrates and invertebrates, and past visual experiences are central to navigational strategies. In *Drosophila*, the behavioral consequences of serotonergic manipulation in visual information processing remain unclear. However, it has been shown that serotonin, via activation of 5-HT2B receptors, modulates the activity of interneurons responsible for early visual processing [39]. Additionally, it is possible to hypothesize that serotonin modulates the behavioral response to visual cues. This would occur via the PLP serotonergic cluster, which innervates and modulates the visual projection neurons (VPNs) that connect early visual processing and higher brain regions [95]. One particular class of VPN, LPLC1, is responsible for tracking visual stimuli [95,96], and could mediate the effects of serotonin on this particular visuomotor response. 

Spatial memory includes learning the visuospatial configurations of environments and using visual information to recall previously encountered locations. In flies, this visual information is relayed from areas such as the OL, the anterior optic tubercle (AOTU), and the bulb (BU), to the EB, where visuospatial memories are generated [86]. Serotonergic neurons can modulate visuospatial memory processing by connecting with cholinergic and GABAergic neurons in the BU and EB, respectively [62,87]. *Drosophila* has been used to assess the integration of multiple sensory stimuli and how that process guide navigation in a complex environment. The study of visuospatial memory has been carried out in flies using a visual-thermal arena. In this behavioral paradigm, flies are guided by visual landmarks to identify a safe location in an otherwise aversive environment [97,98,99]. The safe zone is an area in the arena at 25 °C, surrounded by an aversive environment set at 36 °C. A change of the visual pattern produces a “restoration” of behavior, and flies use spatial information to search for the new safe zone. Repeated learning through many trials results in an increase in spatial search strategies using distal visual landmarks [96]. Thus, flies learn to find spatial locations and remember them to guide their navigation with noticeable efficiency. Ofstad et al. [99] showed that flies retain their visuospatial memories for at least two and up to about 8 h. 

It has been shown that visuospatial memory requires subpopulations of ring neurons from the EB [99], which are described as R3d-p-m and R4d [62]. Several studies show that monoaminergic fibers innervate the EB and contribute to behaviors associated to this neuropil. For instance, it has been shown that dopaminergic signaling modulates sleep performance required for spatial memory [100]. In particular, the ExR2 dopamine neurons from cluster PPM3 [62,101] connect with ring neurons that express a D1-like dopamine receptor [102]. These are evidence in favor of the idea that dopamine contributes to spatial memory and possibly navigation. On the other hand, the ring neurons also show high expression of the d5-HT7 receptor [30,32,62], supporting a serotonergic-EB communication [32,53,103,104]. In addition, serotonin ExR3 neurons from the PMPD cluster [12,101] form parallel and out-of-EB connections with R3d and R3p GABAergic neurons [32,62,88]. ExR2 and ExR3 connecting with different subpopulations of GABAergic ring neurons demonstrates the complex interplay of neural networks underlying spatial memory and navigation.

Additionally, the PMPD cluster seems to be involved in retaining non-visual place memory as well [13]. This suggests that distinct neurons from a unique serotonergic cluster can modulate behavioral responses to different sensory cues, which need to be integrated for spatial cognition and memory. Figure 3C summarizes some of the knowledge gathered over the years regarding the modulatory role played by serotonergic circuits on non-visual and visual information that help flies accomplish spatial navigation.

## 4. Conclusions and Future Directions

The first evidence about serotonin’s behavioral and physiological roles was derived mainly from studies in the whole animal. In this context, the consequences of global or ectopic manipulations of serotonergic components in different behaviors were investigated. These studies suggested that serotonin is implicated in modulating a wide range of behavioral responses. Importantly, these studies also argued in favor of the idea that serotonin plays a neuromodulatory role with structural and functional heterogeneity. This principle is supported by the fact that serotonergic axons originate from a small number of neurons, elaborate complex arborizations, and project diffusely to reach many neurons distributed in distinct brain areas. 

Experimental strategies aimed at dissecting neuronal circuits responsible for cognitive abilities are a step forward in refining our understanding on how serotonin contributes to these behaviors. Although the multisensory control of navigation can be studied in many organisms, the fly’s expansive genetic tools and conserved serotonergic heterogeneity make it an excellent model for identifying the basic principles by which serotonergic circuit motifs modulate the integration in spatial navigation, even considering the differences between vertebrates and *Drosophila*. For example, as discussed here, spatial memory in flies requires subpopulations of ring neurons from the EB, in which R3 and R4 show strong innervation of serotonergic fibers. Spatial memory in flies also depends on the activity of the d5-HT7 receptor. In contrast, in vertebrates, spatial memory involves the MEC and hippocampus, which are brain regions also highly innervated by serotonergic axons. Interestingly, is not the 5-HT7 but the 5-HT1A receptor that seems to play a major role in modulating the activity of these brain nuclei. 

At a clinical level, impairments in spatial cognition are highly significant to several neurological disorders such as Alzheimer’s dementia, traumatic brain injury, and schizophrenia. Additionally, the ability to process and remember spatial representations is associated with learning disabilities in children. Thus, the use of an experimental model such as *Drosophila* to learn some of the basic principles governing spatial navigation could be essential in reaching possible therapeutic actions in neurological disorders and learning disabilities.

## Figures and Tables

**Figure 1 ijms-24-04407-f001:**
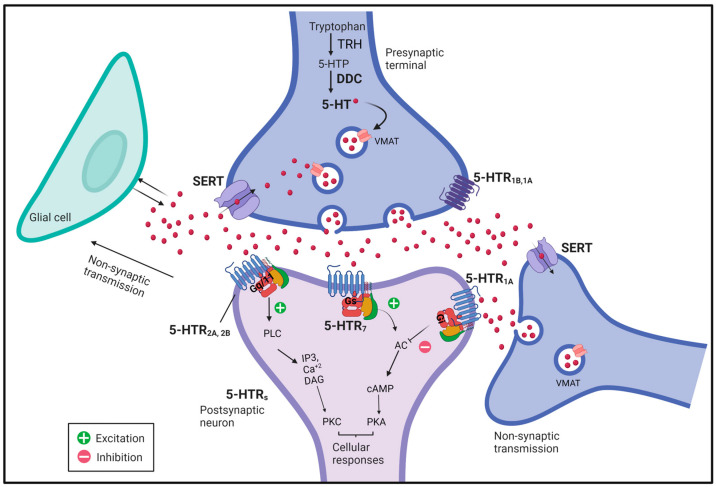
Representative scheme of a serotonergic synapse. The figure shows the distribution of presynaptic and postsynaptic serotonergic components reviewed here. Briefly, the presynaptic region includes components involved in serotonin synthesis, which happens in two steps. First, L-tryptophan is transformed into L-5-hydroxytryptophan (5-HTP) by the tryptophan hydroxylase (dTRH) enzyme. In the second step, 5-HTP is synthesized into 5-hydroxytryptamine (serotonin, 5-HT) by Aromatic L-amino acid decarboxylase (dDDC). The concentration of serotonin in the extracellular milieu is regulated by the activity of the serotonin transporter (dSERT), which is a plasma membrane protein responsible for the reuptake of serotonin back into the presynaptic terminal. Serotonin can then be repackaged into vesicles by the action of VMAT. In the terminal it can be found the d5-HT1B autoreceptor which inhibits the release of serotonin, and the existence of d5-HT1A has also been demonstrated. The postsynaptic region includes five metabotropic receptors for 5-HT: d5-HT1A, d5-HT2A, d5-HT2B, and d5-HT7. The d5-HT1A receptor (orthologous of mammalian 5-HT1A) is coupled to inhibitory Gi/Go, reducing the level of cytosolic cAMP due to the inhibition of the adenylyl cyclase (AC) activity; the d5-HT2A and d5-HT2B receptors (orthologs to mammalian 5-HT2), lead to Ca^+2^ signaling via coupling to Gq, and the activation of the phospholipase C (PLC) pathway; and the d5-HT7 receptor coupled to Gs activates the AC enzyme which leads to the increase in cytosolic cAMP. It has been shown that neuromodulation can occur in aminergic neurons via spillover from the synapse (shown) or via release from non-synaptic sites (not shown) in what is called volume neurotransmission. Glial cells and other neurons contribute to this form of intercellular communication.

**Figure 2 ijms-24-04407-f002:**
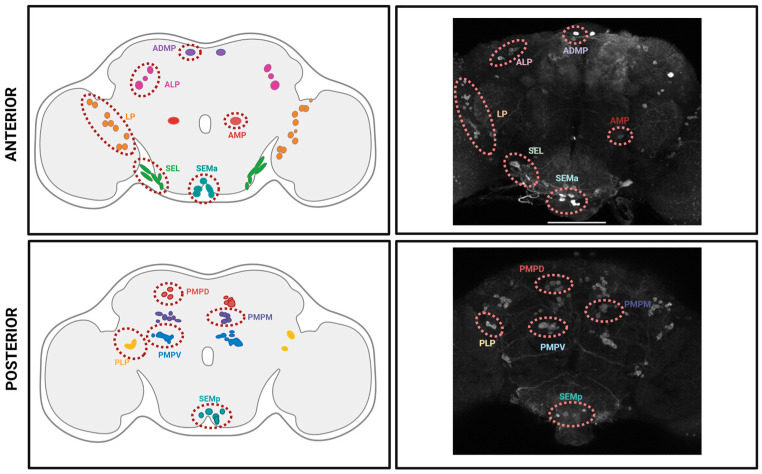
Distribution of serotonergic neuronal populations in the adult fly brain. **Left panels,** schematic illustration of the anterior (top) and posterior (bottom) halves of the adult fly brain, where eleven clusters of serotonin-releasing neurons are identified in red dashed circles, as it follows: ALP, anterior lateral protocerebrum (pink somas); AMP, anterior medial protocerebrum (red somas); ADMP, anterior dorsomedial protocerebrum (in purple); LP (LP2), lateral protocerebrum (orange somas); SEL (SE1–SE2), lateral subesophageal ganglion (green somas); SEM (SE3), medial subesophageal ganglion, which is subdivided in anterior (SEMa, dark cyan somas) and posterior (SEMp, cyan somas) subclusters; PLP, posterior lateral protocerebrum (yellow somas); PMPD, posterior medial protocerebrum, dorsal (light red somas); PMPM, posterior medial protocerebrum, medial (dark purple somas); PMPV, posterior medial protocerebrum, ventral (light blue somas). Schemes are drawn from data and description in [12,58]. **Right panels**, representative images of Trh+-immunoreactive neurons in anterior and posterior halves of a *Drosophila* brain. The serotoninergic clusters are identified in light red dashed circles, and cluster names are indicated. Scale bar: 100 µm. Reprinted from [58].

**Figure 3 ijms-24-04407-f003:**
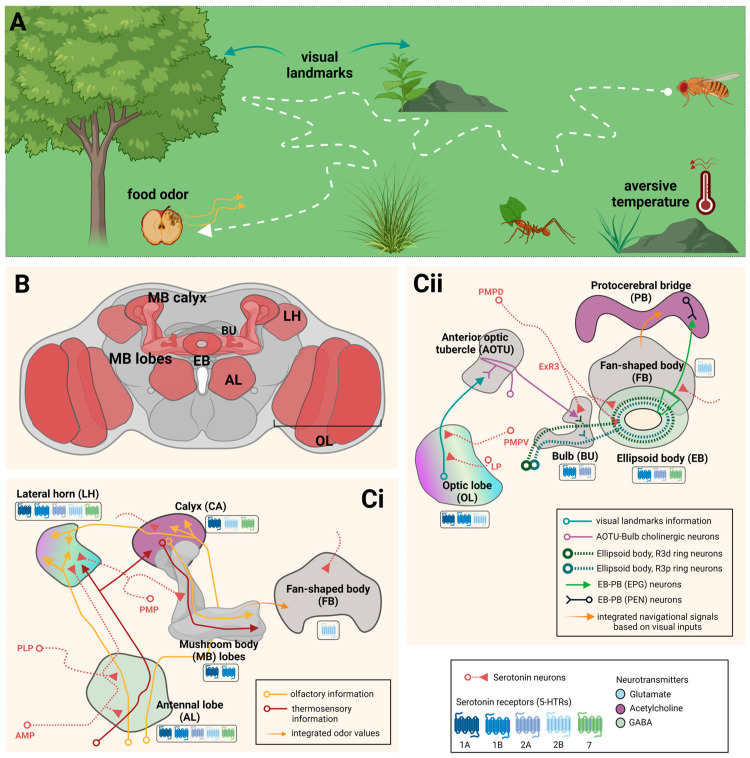
Schematic diagram of neural circuits underlying non-visual- and visual-spatial navigation. (**A**), Flies use navigational strategies involving information provided by different environmental factors, including sensory cues such as food odors (orange arrows), temperature (red arrows), and visual landmarks (dark cyan arrows). (**B**), Scheme showing diverse brain neuropils in the fly brain (in red) that receive and/or process information associated with spatial navigational memories, and are innervated by serotonergic fibers. (**C**), Representation of serotonergic modulation of neural circuits related to sensory and memory processing of non-visual (**Ci**) and visual (**Cii**) information. In boxes are indicated the serotonergic receptors expressed in each fly brain area. Neurons that project to different target areas or that are of different neurochemical identity, are represented in different colors. The graphical schemes were drawn with information published in [61,62,87].

**Table 1 ijms-24-04407-t001:** Historical nomenclature of serotonergic clusters according to the distribution and number ^$^ of neuronal somas ^&^.

Cluster Number	Vallés and White, 1998 [53]	Aleksenyenko et al., 2010 [10]	Pooryasin and Fiala, 2015 [12]	5-HT neurons	Trh-Gal4 > UAS-GFP
**1**	None	ALP	ALP	6 ± 0	4 ± 1
**2**	None	AMP	AMP	2 ± 0	2 ± 0
**3**	None	None	ADMP	2 ± 0	2 ± 0
**4**	LP2a, b	LP2	LP	24 ± 3	18 ± 3
**5**	SE1 and SE2	SE1 and SE2	SEL	10 ± 2	11 ± 1
**6**	SE3	SE3	SEM	10 ± 3	9 ± 3
**7**	SE3	SE3	SEM		
**8**	LP1	PLP	PLP	4 ± 0	4 ± 0
**9**	SP1	PMP	PMPD	6 ± 0	6 ± 0
**10**	SP2	PMP	PMPM	13 ± 2	12 ± 3
**11**	IP	PMP	PMPV	14 ± 3	8 ± 3
**Σ 5HT neurons**				91 ± 13	76 ± 14

^&^ The nomenclature of serotonergic clusters according to published data from [10,12,53]. ^$^ The number of serotoninergic neurons in each identified neuron cluster of the central brain (i.e., excluding the optical lobes) covered by serotonin immunostaining and Trh-Gal4 is indicated as mean ± SD in both hemispheres. The last column shows the description of serotonergic populations when using Trh-Gal4 (Chromosome III, described in [10]) to drive the expression of GFP under the control of UAS.

## Data Availability

Not applicable.

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
