# Peer review of "Navigating Like a Fly: Drosophila melanogaster as a Model to Explore the Contribution of Serotonergic Neurotransmission to Spatial Navigation"

_ijms, 2023, doi:10.3390/ijms24054407_

Round 1

Reviewer 1 Report

In the present manuscript, Gajardo and colleagues attempt to highlight the recent advances in using flies to study the role of serotonergic neurotransmission in spatial navigational memory. The manuscript navigates from the brief introduction to serotonergic system in fruit flies, to involvement of serotonergic transmission in navigational memory.  

1.     5HT1-A could also act as an autoreceptor in addition to 5HT1-B (PMID: 32866139). Unless authors have data to support otherwise authors could add 5HT1-A as a putative autoreceptor, which is expressed presynaptically. 

2.     Authors should provide a reference for “In adult flies, dSERT is widely expressed throughout the brain and is found in cellular bodies, arborizations, and varicosities.”

3.     “Comparably, Drosophila researchers use the Gal4/UAS system [46,47] to express fluorescent proteins and identify the anatomical location of different serotonergic neuron clusters in the adult fly brain.” And “Using this strategy it has been determined that the adult fly brain contains ~100 serotonin-releasing neurons, and 11 neuronal populations can be distinguished per hemisphrere” Researchers mainly use anti serotonin or anti TRH antibody to determine the neuronal identity. The Gal4 driver can be used to drive the expression of gene of interest in the Gal4 labeled cells. The TRH-Gal4 is known to label cells more than the serotonergic neuron (PMID: 26377467). Authors should correct these statements.

4.     “2.2 Anatomical organization of serotonin-releasing neurons and their projections in the adult brain” authors do not elaborate on the projection patterns. Authors could use the recently made available connectome dataset to elaborate on projections or correct the tittle to exclude the word “projections”. 

5.     “3. Serotonergic neural circuits on different aspects of spatial navigation” in this sections authors could provide a brief account on what other systems are participating in the spatial navigation for the reader to appreciate the complexity of the navigation. 

6.     This is optional but authors could consider providing schematics explaining the serotonergic involvement in the spatial memory considering this is the backbone of the review article. 

7.       Are authors aware of any null-mutant (or knockout) studies for genes involved in serotonin homeostasis and signaling that could affect the navigation? 

8.     Authors should check for typo and grammatical errors all over the manuscript, for instance line 9 introduction “similar simlar”.

Author Response

Dear reviewer,

Sincerely,

Jorge Campusano

Reviewer 2 Report

The serotonin system is involved in a wide range of behavioral and neuropsychological processes. Indeed, it is difficult to find basic behavioral responses that are not related to serotonergic signaling.

The serotonergic system is widespread in the animal kingdom, and some of the conserved mechanisms underlying serotonergic modulation of sensory processing have been effectively studied in model organisms (eg, mice, rats, fruit flies, etc.).

In this review, the authors discuss the role of serotonin in spatial navigation, in particular sensorimotor responses and spatial memory processing, using modern data derived from the Drosophila model.

The advantages, possibilities and prospects of using Drosophila for these studies are analyzed competently. The article is well written and interesting to read.

I have only a few minor remarks and recommendations.

*The authors wrote: «In Drosophila, expression binary systems (Gal4/UAS, LexA-LexAop, or Q system) [ 55] and advanced genetic tools (some reviewed in [56]) have helped elucidate the involvement of defined serotonergic neurons and specific neural circuits in modulating behaviors at different levels of complexity.»  Were lines from the Janelia GAL4 collection (collections of driver lines VT GAL4, LexA and split-GAL4 for targeted expression in the Drosophila nervous system) used in the work? Was this genetic tool useful?

* In recent years, CRISPR-Cas gene editing has been actively used to generate mutants in Drosophila. Is there any positive experience of using this system to obtain new mutations that alter serotonergic neurotransmission.

*The text contains some over- long sentences and complex phrases.

«Thus, if we compare the serotonergic system of mammals and Drosophila, it is possible to describe in the fly genome genes encoding presynaptic cellular components such as the biosynthetic enzymes tryptophan hydroxylase, dTRH [18] and dopa decarboxylase, dDDC, which is also known as Aromatic L-amino acid decarboxylase [ 19,20], the vesicular monoamine transporter dVMAT [21], and also the plasma membrane serotonin transporter dSERT, which is responsible for amine transport back into the presynaptic terminal [22]."

“Although the multisensory control of navigation can be studied in many organisms, the fly's expansive genetic tools and conserved serotonergic heterogeneity make it an excellent model for identifying the basic principles by which serotonergic circuit motifs modulate the integration in spatial navigation, even considering that the serotonergic target areas between vertebrates and Drosophila are not conserved.”

I recommend modifying such sentences because they are difficult to understand in this form.

Author Response

Dear reviewer,

Jorge Campusano.

Round 2

Reviewer 1 Report

I do not have any further concerns. Authors have addressed my queries.